# Engineering Possibility Studies of a Novel Cylinder-Type FOWT Using Torus Structure with Annular Flow

**Xiaolei Liu** [1,*] and **Motohiko Murai** [2,*]

1 Graduate School of Engineering Science, Yokohama National University, Yokohama 2400067, Japan
2 Graduate School of Environment and Information Sciences, Yokohama National University, Yokohama 2400067, Japan
* Correspondence: liu-xiaolei-tk@ynu.jp (X.L.); m-murai@ynu.ac.jp (M.M.)

**Abstract:** This paper proposes and researches a novel cylinder-type FOWT using a neutrally buoyant double-layer torus structure with annular flow; its oscillatory motion in severe sea conditions is controlled by a spinning top device designed as a neutrally buoyant double-layer torus structure with annular flow water in a torus structure with a small internal radius, and welded to the periphery of the cylinder-type FOWT underwater buoyancy-providing part. The rotational axis retention effect and the gyroscopic effect are considered appropriate approaches to suppress the oscillating motion of FOWT. To obtain a better hydrodynamic response, the scale of the torus structure, such as its radius, the radius of the internal annular flow water, and the angular velocity of the annular flow water are taken as the design parameters, and a large number of comparative calculations based on the fluid–solid coupling theory of potential flow are carried out to determine the appropriate design parameters. Eventually, on the basis of the obtained suitable design parameters, the proposed conceptual design approach is demonstrated to be feasible in view of the energy consumption.

**Keywords:** spinning top; annular flow; torus structure; hydrodynamic response; design parameters

## 1. Introduction

In the context of global warming, carbon dioxide peaking and carbon neutrality have become goals that we are striving to achieve. Ocean wind energy has always a renewable energy source valued by people, used from ancient sailing ships to modern ocean wind power generation. Up to 2020, fixed offshore wind turbines were developed for 35 GW power [1]; however, when the water depth exceeds 50–60 m, conventional fixed offshore wind turbines are no longer economical. In response to this problem, Professor William Edward Heronemus of the University of Massachusetts at Amherst (UMass) put forward the concept of an FOWT for the first time in 1972 [2]. From then on, a scale-model pool test phase in the 1990s, a low-power prototype test stage in the 2000s, and a megawatt unit prototype demonstration stage in the 2010s were presented. At present, some commercial projects have been carried out, but only ~65.7 MW FOWTs have been built worldwide [3].

There are mainly four types of FOWTs from the perspective of the foundation supporting structure: spar-type platform [4], semi-submersible platform [5], barge-type platform [6], and tension leg platform (TLP) [7]. Each has its advantages and disadvantages. Nevertheless, because of their high center of gravity, the hydrodynamic response in the roll and pitch DOFs are relatively large under severe sea conditions. Therefore, if the vast level of the far-reaching sea wind energy is to be used on a large scale, the problem of restraining the oscillating movement of FOWT that has always plagued us should be researched and resolved seriously. The current methodologies for anti-rolling technology are summarized below.

## 1.1. Increasing the Damping Coefficient

A bilge keel can achieve the anti-rolling by generating and releasing eddy currents, especially at zero speed, but the overall anti-rolling effect is not too big. Furthermore, an anti-rolling tank as a tuned mass damper (TMD) can also achieve anti-rolling by tuning its oscillation frequency to be similar to the resonant frequency of the object it is mounted on. However, it will occupy a lot of space [8].

## 1.2. Changing the Natural Frequency of FOWT

This method keeps the natural frequency of FOWT away from the frequency of the most frequent waves occurring in the surrounding sea area. In addition to adjusting its mass and geometric properties, such as the mooring line, the anti-rolling effect can be achieved by providing horizontal resilience, whereby the traditional catenary mooring line provides horizontal resilience via its weight and the taut mooring line provides horizontal resilience via its elastic tension. This is the most practical method and is widely used in actual production. However, the mooring line requires a firm connection to the seabed, and, when the water depth is super deep, too long of a mooring line is very inconvenient [9].

## 1.3. Reducing the Wave Exciting Force or Moment

In this case, the rudder and the anti-rolling fin can achieve a satisfactory anti-rolling effect but only at a relatively high sailing speed. Moreover, the sensors feed the measured environmental information to a dynamic positioning system (DPS), which then controls propulsion systems such as propellers to generate forces that oppose the environmental forces. However, the DPS as an active control method is a very complex system [10].

## 2. Conceptual Design

This paper proposes and researches a novel cylinder-type FOWT using a neutrally buoyant double-layer torus structure with annular flow. A torus structure is welded to the periphery of the cylinder-type FOWT underwater buoyancy-providing part. As shown in Figure 1, there are three dimensions: the radius of the torus structure is the distance from the center of the ring tube to the center of the torus; the radius of the internal annular flow water is the radius of the ring tube of the internal torus; the radius of the external torus is the radius of the ring tube of the external torus. In this paper, the design parameter of the radius of the torus structure was varied from 35 m to 60 m in 5 m intervals; the design parameter of the radius of the internal annular flow water was varied from 1 m to 6 m in 1 m intervals.

The torus structure is very common in daily life, such as in swimming rings, doughnut desserts, and bicycle tires. In offshore engineering, a torus structure floating on the water is placed on the periphery of the FOWT to act as a cushion or wave energy converter [11,12]. The torus structure is also often used in scientific research, such as the acceleration track of a particle accelerator in particle physics. The torus section in this article has a double-layer structure with a large external torus and a small internal torus in the large external torus, and these two parts are joined together by welding some steel rods. The large external torus and the small internal torus have the same center of the ring tube and the same center of the torus. All space in the ring tube of the internal torus is densely filled with fluid. The radius of the ring tube of the external torus is determined so that it has neutral buoyancy to keep a constant draft of FOWT for comparative studies. Moreover, the neutral buoyancy can also facilitate installation and removal. If this idea is used in actual production, to maintain the FOWT on its site, a mooring system should be installed or multi-spinning top devices will be needed, similar to a helicopter using two rotors to hover. In addition, the structural and fatigue assessments of the whole FOWT hull should be further studied.

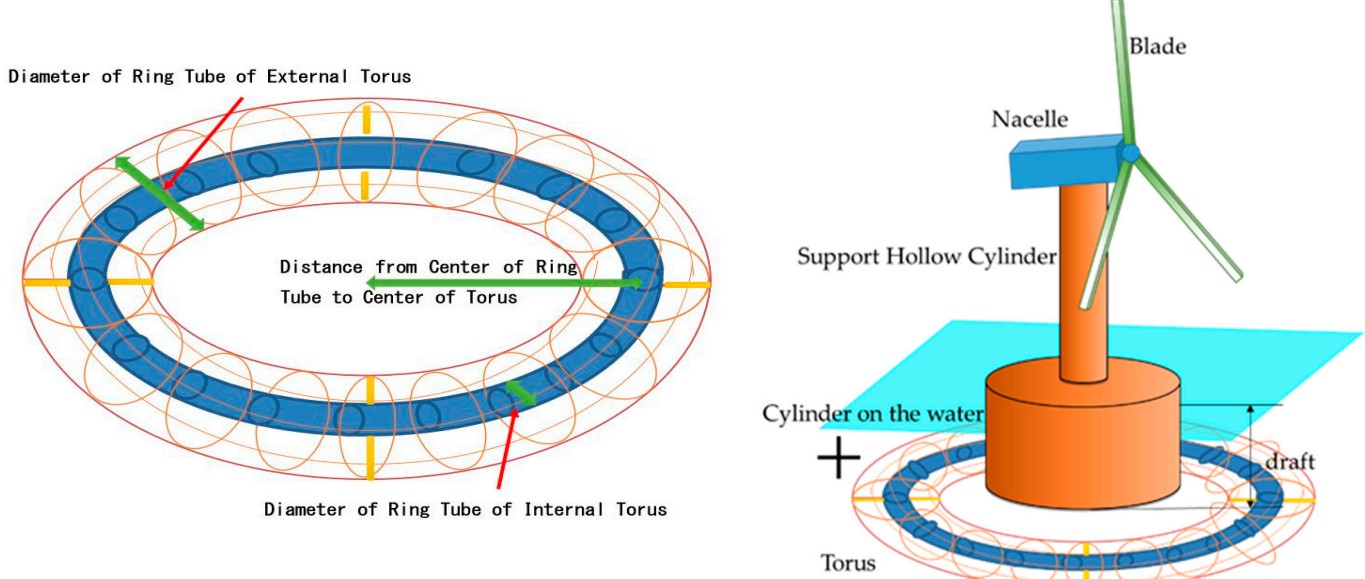

**Figure 1.** The designed two-layer torus structure and cylinder-type FOWT.

It was envisaged [13] that the principle of the rotating annular flow in the small internal torus is the same as the rotation of a rigid body, i.e., the spinning top, as experimentally verified in Section 4.1. The fact that the Moon revolves around the Earth without resistance every lunar month plays a vital role in the stability of the Earth's rotation axis. The tank's gun barrel uses a gyro stabilizer to allow it to fire accurately even when driving on bumpy roads. In addition to being used as a stabilizer, the spinning top is also widely used in many fields of engineering. For example, it was used to develop a gyrocompass for navigation 100 years ago; NASA envisaged using it to create gravity in space vehicles, but the rotation radius of the space vehicles was required to exceed 100 m [14]; furthermore, a three-axis gyroscope as a microelectromechanical system was introduced into the iPhone 4 released in 2010. However, the torus itself does not need to make a rotational movement in the presented case; the water in the small internal torus is assumed to flow evenly, and the gyroscopic effect from the precession of annular flow and the rotational axis retention effect from the rotational inertia of annular flow can be obtained to actively restrain the oscillating movement of the FOWT. If these effects are linearized into a small-amplitude problem, they can be treated as a damping force according to the derivation presented in the next section. Indeed, the space between the external torus structure and central cylinder will be the same as a moon pool. The water in this space struggles to form progress waves and will have impact on the heave, roll, and pitch DOFs. Because this space is not a large volume, its impact is ignored in this paper. This paper studies the possibility of contributing to a reduction in the oscillatory motion of a cylinder-type FOWT and improving the hydrodynamic response by effectively using these effects. The presence or absence of the torus structure, the different positions placed in the vertical direction, the radius of the torus structure, the radius of the internal annular flow, the angular velocity of the annular flow, the central cylinder radius, the central cylinder height, and the central cylinder wall thickness were taken as the design variables.

## 3. Formulation

To restrain the oscillating movement of an FOWT, predicting the movement of the FOWT under the action of water waves is necessary. In physics, the different phenomena can be highly integrated, representing different sides of the same coin. The method of analyzing the fluid–solid coupling problem based on potential flow is similar to the method of analyzing reflection and refraction phenomena in fluctuating optics, but hydrodynamics has more complex aspects, especially with respect to hydrodynamic parameters such as

the added mass and complex geometry of wet surfaces. Eventually, in order to obtain the various design quantities of an FOWT floating on random ocean waves, data on the spectral distribution that can stochastically express ocean waves and on the frequency response in the regular waves of FOWT are first required, and then the principle of linear superposition and the maximum prediction theory for Longuet–Higgins statistics can be applied [15].

### 3.1. Hydrodynamic Mathematical Model and The Motion Equation

As shown in Figure 2, the seakeeping problem is seen as a linear system, the output response achieves a steady state after enough time of a harmonic steady frequency $\omega$ oscillating input. The motion equation of a floating body in the frequency domain is shown in Equation (1).

$$\sum_{j,i=1}^{6} \left\{ -\omega^2 (m_{ji} + \mu_{ji}) - i\omega\lambda_{ji} + C_{ji} \right\} x_i = f_j, \tag{1}$$

where $m_{ji}$ is the mass in the translational movement and moment of inertia in the rotational movement, $\mu_{ji}$ is the added mass, also called Newton's zeroth law in which the additivity of mass is caused by the inertia force because of the movement of the surrounding flow field of the floating body, $\lambda_{ji}$ is the damping coefficient, $C_{ji}$ is the resilience coefficient, and $f_j$ is the exciting wave force consisting of the Froude–Krylov force and the diffraction force [16].

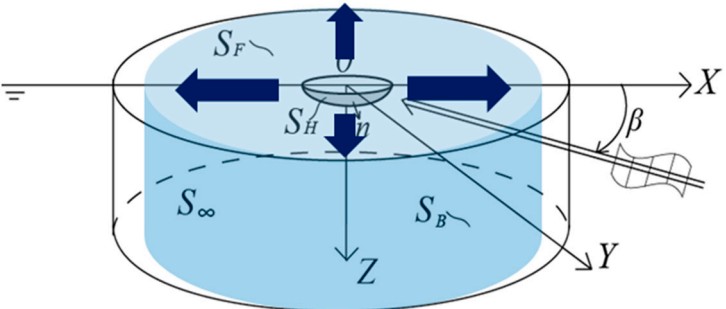

**Figure 2.** The hydrodynamic mathematical model.

### 3.2. The Spinning Top

As shown in Figure 3, when the rigid body rotates, it should have an axial retention effect and a gyroscopic effect from the conservation principle of angular momentum. As the conservation law of angular momentum applies only in an inertial coordinate system, the angular momentum $\mathbf{L}$ of a rotating rigid body can be changed from a fixed coordinate system (an inertial coordinate system) to a rotating coordinate system (a non-inertial coordinate system), and then the motion equation of a rotating rigid body can be derived as follows:

$$\mathbf{M} = \frac{d\mathbf{L}}{dt} = J\frac{d\boldsymbol{\omega}(t)}{dt} = J\frac{d^*\boldsymbol{\omega}}{dt} + \boldsymbol{\omega} \times (J \cdot \boldsymbol{\omega}), \tag{2}$$

where $\mathbf{M}$ is moment, $J$ is moment of inertia, $\frac{d^*\omega}{dt} = \frac{d\omega_x}{dt}\hat{x} + \frac{d\omega_y}{dt}\hat{y} + \frac{d\omega_z}{dt}\hat{z}$ is the angular acceleration in the rotating coordinate system, and $\frac{d\omega}{dt} = \frac{d}{dt}(\omega_x\hat{x} + \omega_y\hat{y} + \omega_z\hat{z})$ is the angular acceleration in the fixed coordinate system. Eventually, an object moves at a velocity $v$ relative to the inertial coordinate system, and the coordinate system in which the object is located moves relative to the inertial coordinate system with an angular velocity $\boldsymbol{\omega}$. Accordingly, the acceleration relative to the inertial coordinate system is $\mathbf{a} = \frac{d^2\mathbf{r}}{dt^2} + 2\boldsymbol{\omega} \times \frac{d\mathbf{r}}{dt} + \boldsymbol{\omega} \times (\boldsymbol{\omega} \times \mathbf{r})$. The responses of the pitch, roll, and yaw DOFs are expressed in Equations (3) and (4) within the range of linear theory.

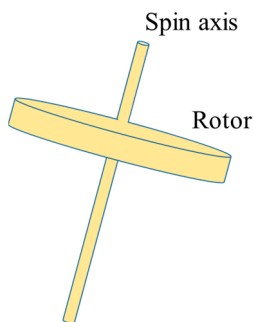

**Figure 3.** The spinning top.

In this way, we can acquire the axial retention effect coefficient matrix $N_{keep}$ and the gyroscopic effect coefficient matrix $N_{gyro}$. The gyroscopic moment is obtained from the precession motion, a phenomenon in which the rotation axis of a rotating rigid body rotates around another fixed axis, similar to the Coriolis force calculated by $F_c = -2m(\boldsymbol{\omega} \times \boldsymbol{v})$ [17]. In this way, the axial retention effect and the gyroscopic effect can be recognized as a damping term proportional to the angular velocity of vibration by linearizing. Incorporating these effects of annular flow into Equation (1), the motion equation of a floating body with a rotating rigid body in the frequency domain can be expressed as Equation (5).

$$\begin{bmatrix} M_{4keep} \\ M_{5keep} \\ M_{6keep} \end{bmatrix} = -i\omega N_{keep} \begin{bmatrix} x_4 \\ x_5 \\ x_6 \end{bmatrix} = -i\omega \begin{bmatrix} J_{yy}\omega_{yr} + J_{zz}\omega_{zr} & 0 & 0 \\ 0 & J_{xx}\omega_{xr} + J_{zz}\omega_{zr} & 0 \\ 0 & 0 & J_{xx}\omega_{xr} + J_{yy}\omega_{yr} \end{bmatrix} \begin{bmatrix} x_4 \\ x_5 \\ x_6 \end{bmatrix}. \tag{3}$$

$$\begin{bmatrix} M_{4gyro} \\ M_{5gyro} \\ M_{6gyro} \end{bmatrix} = -i\omega N_{gyro} \begin{bmatrix} x_4 \\ x_5 \\ x_6 \end{bmatrix} = -i\omega \begin{bmatrix} 0 & (J_{zz} - J_{yy})\omega_{zr} & (J_{zz} - J_{yy})\omega_{yr} \\ (J_{xx} - J_{zz})\omega_{zr} & 0 & (J_{xx} - J_{zz})\omega_{xr} \\ (J_{yy} - J_{xx})\omega_{yr} & (J_{yy} - J_{xx})\omega_{xr} & 0 \end{bmatrix} \begin{bmatrix} x_4 \\ x_5 \\ x_6 \end{bmatrix}. \tag{4}$$

$$\sum_{j,i=1}^{6} \{-\omega^2 (m_{ji} + \mu_{ji}) - i\omega(\lambda_{ji} + N_{keepji} + N_{gyroji}) + C_{ji}\}x_i = f_j. \tag{5}$$

### 3.3. The Expected Value during Irregular Waves

By making use of the normalized RAO of regular waves acquired in Section 3.2, the expected value of the hydrodynamic response can be calculated during an irregular wave. An irregular wave is defined through the wave spectrum $S_\zeta(\omega)$, and it can be expressed as a superposition of regular waves. The correlation function of the irregular wave function reflects the distribution state of wave energy, and the integrated area of the spectral density function $S_{\zeta\zeta}(\omega)$ is the average wave energy. The correlation function and the spectral density function exhibit a Fourier transform relationship. The expected value $\theta_{pitch}$ can be calculated from the integrated amount of the floating body response spectrum $S_z(\omega)$ [18].

$$E\left[\zeta^2(x, y, t)\right] = \int_0^\infty S_\zeta(\omega)d\omega. \tag{6}$$

$$S_{\zeta\zeta}(\omega) = \frac{2}{\pi} \int_0^\infty R_{\zeta\zeta}(\tau) \cos(\omega\tau)d\tau. \tag{7}$$

$$S_z(\omega) = |RAO(\omega)|^2 S_{\zeta\zeta}(\omega). \tag{8}$$

$$\theta_{pitch} = \sqrt{2 \int_0^\infty S_z(\omega)d\omega}. \tag{9}$$

The wave spectrum in this calculation is the Pierson–Moskowitz (P–M) spectrum of Equation (10) [19].

$$\Phi_{\zeta\zeta}(\omega) = \frac{\alpha g^2}{\omega^5} e^{-\beta\left(\frac{g}{\omega V}\right)^4}, \tag{10}$$

where $\alpha = 8.10 \times 10^{-3}$, $\beta = 0.74$, $g$ is the gravitational acceleration, and $V$ is the wind speed at 19.5 m above sea level [20]. Therefore, once we take some important parameters of a floating body as the input data of the above mathematical model, the output data of the hydrodynamic response of the floating body in the ocean can be obtained through numerical calculation.

## 4. Results and Discussion

### 4.1. Experimental Verification

To verify the principle of the motion equation of the floating body in consideration of the annular flow, a simple pendulum experiment, as shown in Table 1 and Figure 4, was carried out. The tube was wrapped around a disc in three circles, and the water flow speed inside the tube was controlled by a Masterflex® L/S® pump machine (Masterflex, Vernon Hills, IL, USA), while a ZMP IMU-Z six-axis motion sensor equipped with a three-axis accelerometer and three-axis gyro sensor was used to measure the vibration signal, according to the equation of motion of a simple pendulum.

$$ml^2\frac{\mathrm{d}^2\theta}{\mathrm{d}t^2} = -mgl\sin\theta + \beta\frac{\mathrm{d}\theta}{\mathrm{d}t} \text{ the initial conditions are } \theta\mid_{t=0}=\theta_0, \frac{\mathrm{d}\theta}{\mathrm{d}t}\mid_{t=0}=\dot{\theta}_0 = 0 \quad (11)$$

When the amplitude $\theta$ is a small angle and less than $10°$, $\sin\theta$ can be linearized into $\theta$ via Taylor expansion of $\sin\theta$; thus, the equation of motion can be simplified to

$$\frac{\mathrm{d}^2\theta}{\mathrm{d}t^2} + 2n\frac{\mathrm{d}\theta}{\mathrm{d}t} + k^2\theta = 0 \; n = \frac{-\beta}{2ml^2}, \; k = \sqrt{\frac{g}{l}}, \quad (12)$$

where the damping ratio $\xi = \frac{n}{k}$. Four independent experiments were performed by proportionally varying the water flow speed $\omega_{water}$ inside the tube. Through the $\theta$–$t$ curve as shown in Figure 5 measured by the six-axis motion sensor, the size of the damping ratio $\xi$, as shown in Table 2 and Figure 6, could be identified. A linear increase in the damping ratio $\xi$ with the angular velocity of the annular flow water can be used to verify the relationship between the rotating annular flow in the torus structure and the axial retention effect.

**Table 1.** The main parameters of the experimental equipment.

| Section | Item | Size (mm) | Weight (g) |
|---|---|---|---|
| **Bar** | Length | 900 | 46.5 |
| | Width | 30 | |
| | Height | 10 | |
| **Disc** | Radius | 120 | 14 |
| | Radius of central hollow part | 17.5 | |
| | Height | 25 | |
| **Sensor** | Length | 15 | 1.7 |
| | Width | 10 | |
| | Height | 10 | |
| **Tube** | Outer radius | 3.5 | 175 |
| | Inner radius | 3 | |
| | Length | 3580 | |
| **Annular water** | Length | 2639 | 76 |

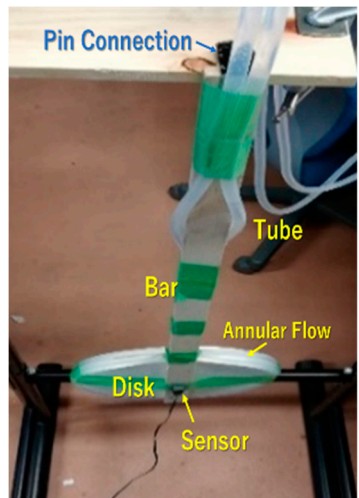
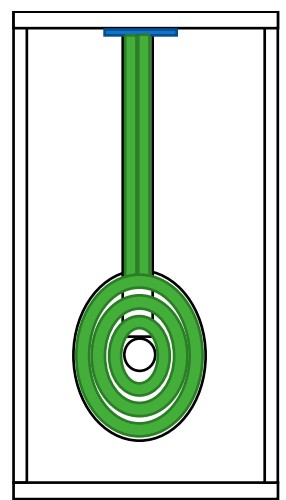
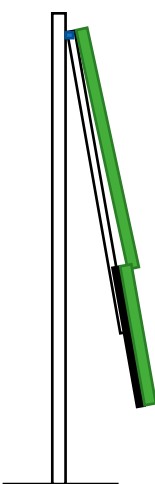

**Figure 4.** Experimental setup, with front and side schematic views.

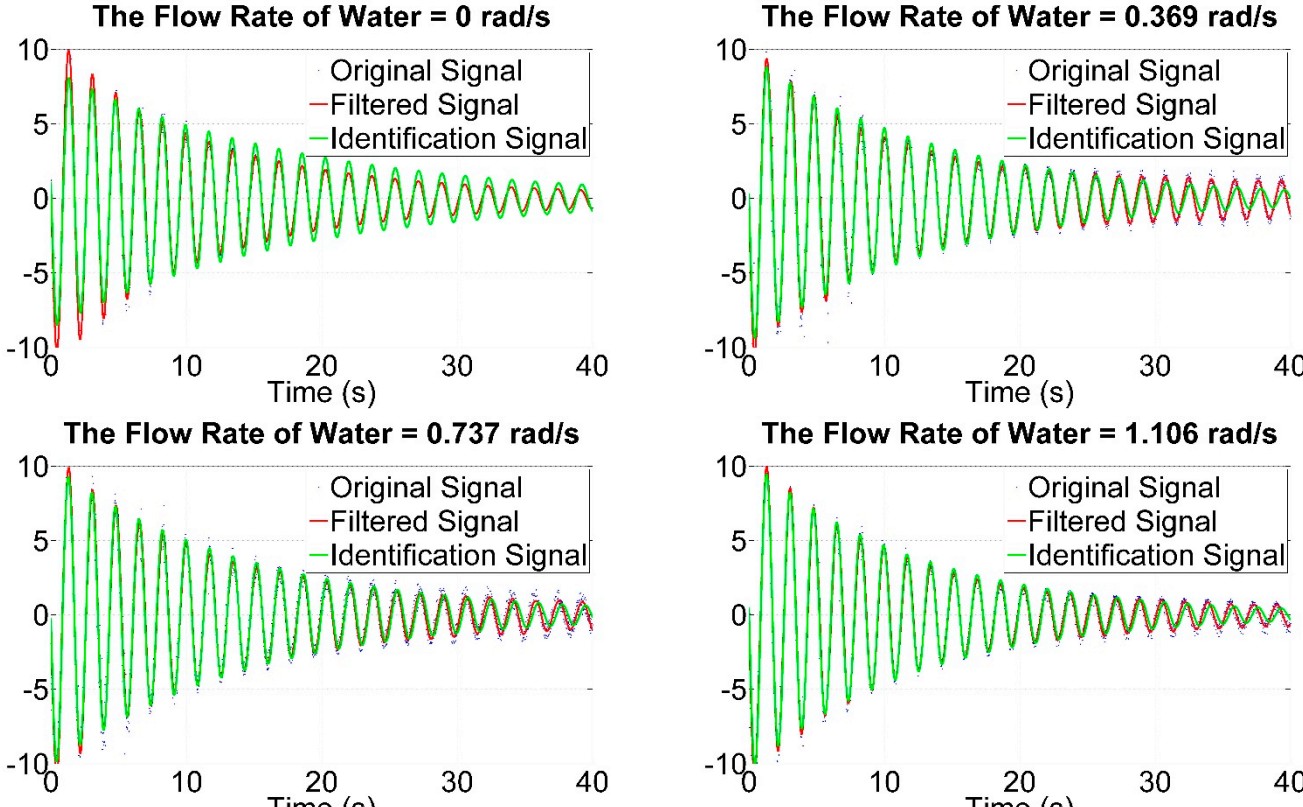

**Figure 5.** Experimentally measured signal curves and fitted curves.

**Table 2.** Identified damping ratio result.

| No. | Initial Tilt Angle (°) | $\omega_{water}$(rad/s) | Damping Ratio $\xi$ (%) |
|---|---|---|---|
| 1 | 10 | 0 | 1.9220 |
| 2 | 10 | 0.369 | 2.0817 |
| 3 | 10 | 0.737 | 2.1223 |
| 4 | 10 | 1.106 | 2.3373 |

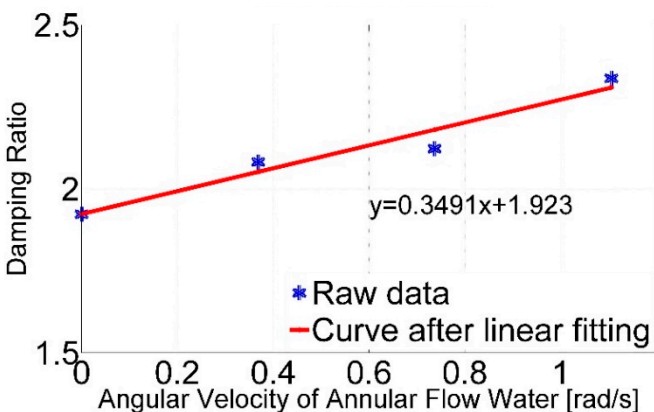

**Figure 6.** Curve after linear fitting.

Because the obtained vibration curves featured many burrs and were not smooth, the multipoint averaging and smoothing method in the time domain and a high-pass filter in the frequency domain, with the cutoff frequency from 2 Hz to 50 Hz, were used to eliminate the high-frequency noise and to improve the signal-to-noise ratio of the vibration signal. Then, modal parameter identification was carried out using a complex exponential method.

According to the experimental data, the damping ratio $\xi$ increased linearly with the angular velocity of the annular flow water, thus confirming the axial retention effect, which acted on the damping effect of the simple pendulum. Consequently, the validity of the motion equation of the torus structure with annular flow shown in Equation (3) was confirmed.

*4.2. The Object Parameters*

Generally speaking, the capacity factor of onshore wind farms is 30%, while, for offshore wind turbines, 40% or more is desirable. To achieve such a high capacity factor, an annual average wind speed of about 7.5 m/s or more is needed [21]. At present, the research on the power generation efficiency of horizontal-axis wind turbines is based on the blade element momentum (BEM) theory [22], which can calculate the wind energy that horizontal-axis wind turbines can absorb from the flowing air and wind load acting on the blades. According to the BEM theory, the wind energy utilization factor is calculated as

$$C_p = \frac{2\rho A_d v_\infty^3 a(1-a)^2}{0.5\rho A_d v_\infty^3} = 4a(1-a)^2. \tag{13}$$

Therefore, $dC_p/da = 12a^2 - 16a + 4$, and, when $a = 1/3$, the wind energy utilization factor $C_p$ becomes the maximum value of 59.26%. This is also called the Betz limit [23]. At this time, the energy power absorbed by the blade plane from the airflow is

$$P_{max} = \frac{8}{27}\rho A_d v_\infty^3, \tag{14}$$

where $P_{max}$ is the maximum absorbed wind energy power generating electrical energy power, $\rho$ is the air density, $A_d$ is the sweeping area when the blades rotate, and $v_\infty$ is the wind velocity in the upwind side [24]. According to dimensional analysis, this formula is also correct.

In actual production, with the capacity factor as the design goal, the rated wind speed, rated output power, and size parameters of the wind turbine can be determined on the basis of local meteorological data. This paper selected an FOWT with a rated output power of 20 MW and a rated wind speed of 12 m/s as the research object. The parameters of the FOWT are presented in Table 3.

**Table 3.** The parameters of FOWT.

| Parameter | Units (m) | Parameter | Units (m) |
|---|---|---|---|
| Single blade length | 100 | Nacelle width | 10 |
| Single blade width | 1 | Nacelle height | 8 |
| Single blade wall thickness | 0.2 | Nacelle wall thickness | 0.8 |
| Support hollow cylinder radius | 5 | Radius of cylinder in water | 30 |
| Support hollow cylinder length | 110 | Height of cylinder in water | 45 |
| Support hollow cylinder wall thickness | 0.3 | Wall thickness of cylinder in water | 0.15 |
| Nacelle (hollow but thick wall) length | 30 | Ballast height | 12 |

*4.3. The Parametric Study of FOWT under the Water Plane*

All FOWTs were assumed to be floating in the ocean with a water depth of 200 m, where a change in the wave period T from 1 s to 15 s indicated a deep water wave. Regular waves were propagated in the direction of the pitch DOF when performing the simulation calculation. The laboratory's existing in-house fluid–solid coupling hydrodynamic CAE program "SS (Three-Dimensional Sink–Source Method)" [25,26] was used, where the number of nodes of all models was 2581, the number of elements of all models was 2565, and the elements were quadrilateral panels. Other calculations were completed using MATLAB R2016b [27].

4.3.1. The Presence or Absence of the Torus Structure and the Different Positions Placed in the Vertical Direction

In the Figure 7, M40-7 refers to the model of the radius of the torus structure, where the distance from the center of the tube to the center of the torus was 40 m, and the ring tube radius of the internal small torus was 7 m.

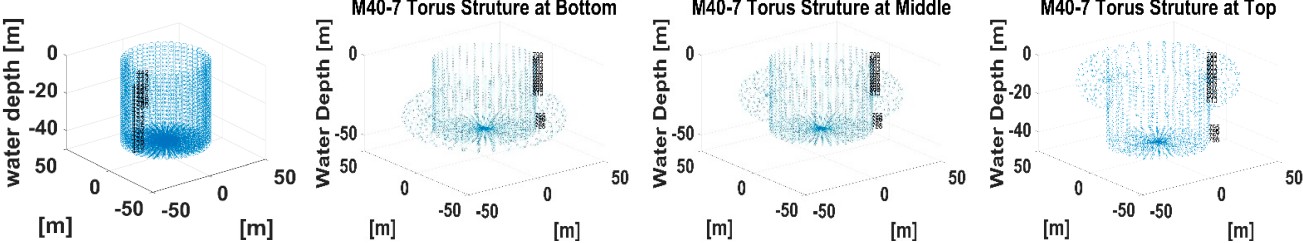

**Figure 7.** The wet surface mesh of M40-7.

As can be seen from Figure 8, for an FOWT with a torus structure, a bigger natural period can be obtained than without a torus structure, because the former has a larger mass coefficient and added mass coefficient, but the resilience coefficient does not change. However, during actual sea wave periods, there is almost no effect on the surge and pitch DOFs. The torus structure at the top is expected to withstand more intense waves, which is beneficial to the overall hydrodynamic response, but this will increase the overall center of gravity and buoyancy, as well as reduce the moment of inertia around the horizontal axis, which will be harmful to the overall hydrodynamic response. However, the torus structure at the top does not have a better hydrodynamic response than the torus structure in middle, thereby offsetting these advantages and disadvantages. When the torus structure at the bottom has the best hydrodynamic response, the advantages of lowering the overall center of gravity and buoyancy can have a greater impact than the disadvantages.

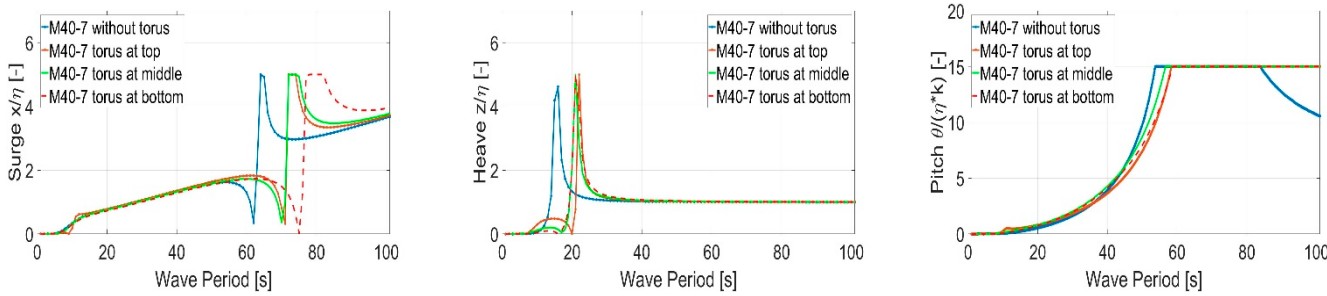

**Figure 8.** The normalized RAO of surge, heave, and pitch DOFs of M40-7.

### 4.3.2. The Radius of the Torus Structure

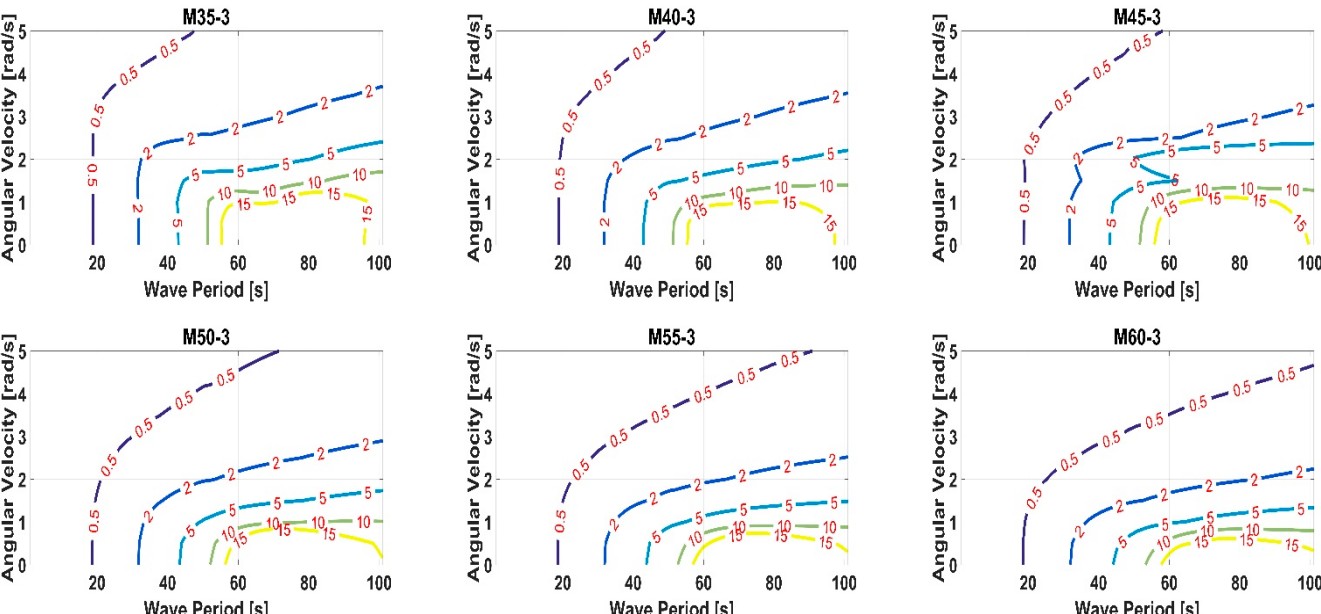

**Figure 9.** The contour graphs of the normalized RAO in the pitch DOF for torus structures with different radii.

In the irregular wave calculation, as shown in Figure 10, the Beaufort wind scale was assumed to be at level 10 (gale), with a significant wave height of 9 m and average wave period of 20 s. This is because the natural period of the designed FOWT in this paper is far greater than the actual sea wave period; thus, the selected average wave period was a little bigger than the offshore structures used, and greater expected pitch values could be achieved, reflecting the simulation calculation in extremely hazardous situations.

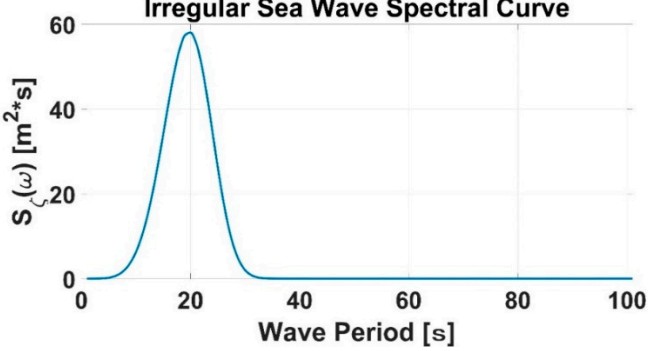

**Figure 10.** The irregular sea wave spectral curve.

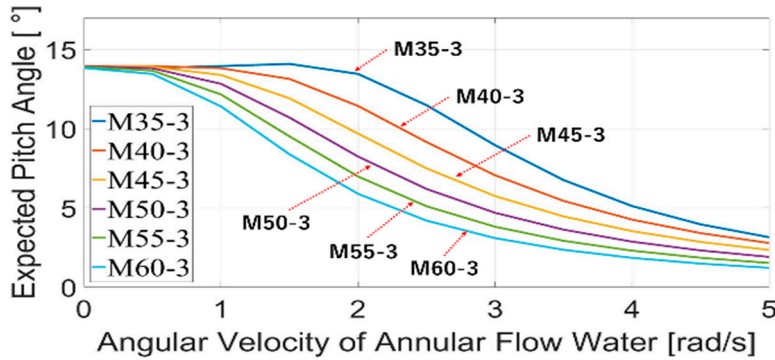

**Figure 11.** The expected pitch angle in the irregular wave for torus structures with different radii.

4.3.3. The Radius of the Internal Annular Flow

In the case of the above two kinds of design variables, it can be confirmed that the normalized RAO of the pitch DOF of the FOWT was substantially attenuated near the natural period with the increase in the angular velocity of annular flow, and a large damping effect could be obtained, even at a relatively small angular velocity. In particular, when the angular velocity was less than 2 rad/s, the radius of the torus structure was over 40 m, and the radius of the ring tube of the small torus was over 3 m, as shown for M40-3 in Figures 9 and 12, the dimensionless pitch DOF response value was suppressed to 5 or less in a fairly wide periodic band. Therefore, a large damping effect could be demonstrated even at a relatively small angular velocity. In addition, the direction of the contour lines was inclined parallelly to the angular velocity, signifying that the increase in angular velocity did not have much of an effect when the angular velocity exceeded a certain value.

By looking at the response of the irregular wave in Figures 11 and 13, we can see that the expected value was suppressed to less than half at an angular velocity of 3 rad/s when the radius of the ring tube of the internal torus was over 3 m. However, in the case of M40-1, the proportion of the moment of inertia of water $J_y$ was less than 1%, and the oscillating suppression effect could not be expected despite the tremendous increase in the angular velocity. Moreover, in the case of M40-2, the proportion of the moment of inertia of water $J_y$ was less than 3%, and the oscillating suppression effect could again be hardly expected. In addition, as the proportion of the moment of inertia of water $J_y$ increased, the oscillating suppression effect not only increased linearly but the growth trend also became increasingly smaller. Ultimately, this indicated that the oscillating suppression effect could be expected even for a relatively small angular velocity by setting the proportion of the moment of inertia of water $J_y$ to 5% or more.

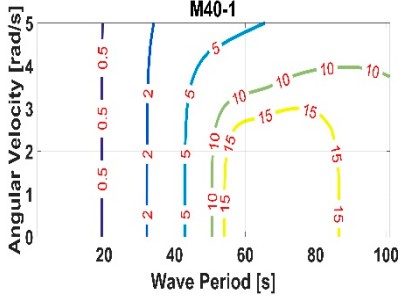
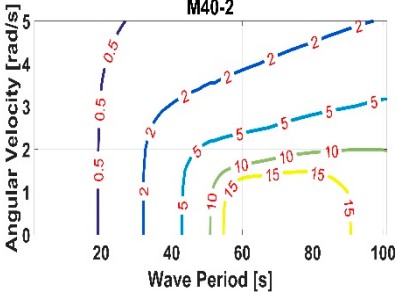
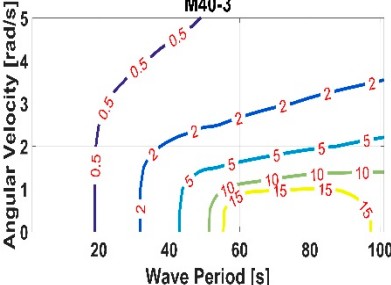

**Figure 12.** *Cont.*

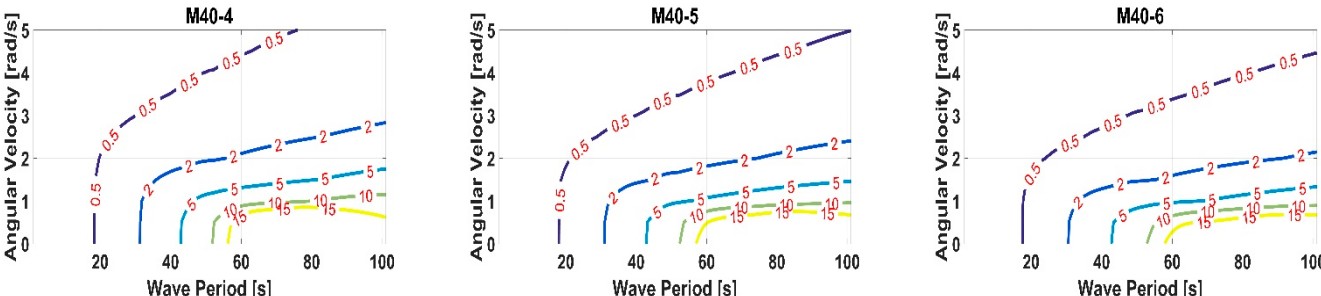

**Figure 12.** The contour graphs of the normalized RAO in the pitch DOF upon changing the internal annular flow radius.

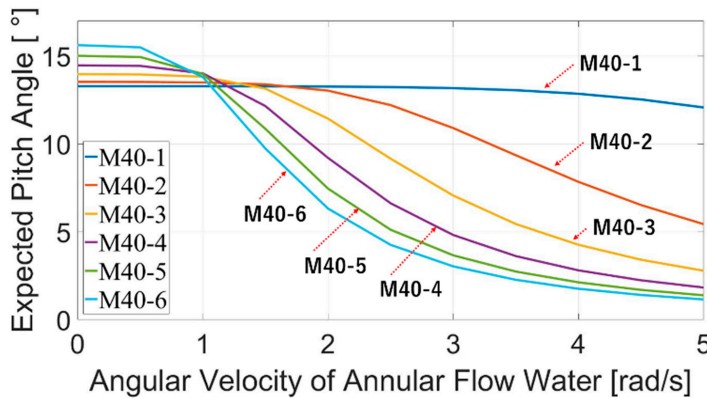

**Figure 13.** The expected pitch angle in the irregular wave upon changing the internal annular flow radius.

### 4.3.4. The Central Cylinder Radius and Height

From Figure 14, we can see that, as the central cylinder radius was increased from 28 m to 32 m, when the angular velocity of annular flow was relatively low, the hydrodynamic performance worsened significantly. At this time, although the GM was increased from 0.15 to 4.35 m, the water plane area also increased from 2463 to 3217 m², which led to a reduction in the natural period of the FOWT from over 100 s to about 40 s, which is very close to the average wave period of the irregular wave. Accordingly, the expected value become larger, reaching 28°, which is far greater than the limit value used in the FOWT design. When the angular velocity of the annular flow reached 3 rad/s, the hydrodynamic response tended to be the same, because the proportion of the moment of inertia Jy occupied by the water part of the annular flow was only slightly reduced from 5.54 to 4.57%.

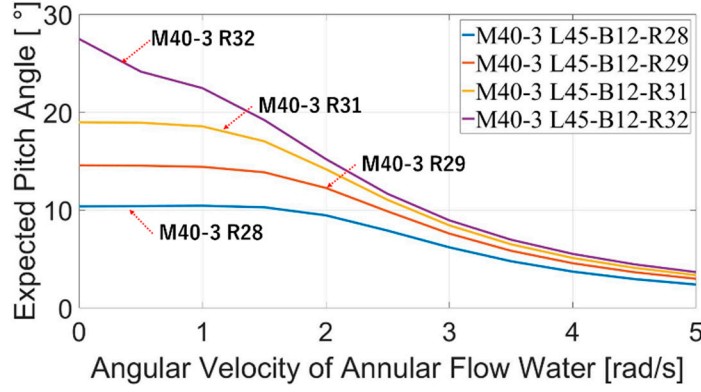

**Figure 14.** The expected pitch angle in the irregular wave upon changing the central cylinder radius.

From Figure 15, we can see that, as the central cylinder height was increased from 43 m to 47 m, a similar hydrodynamic performance was obtained regardless of the angular velocity of annular flow, since the GM only changed from 2.39 to 2.19 m, and the proportion of the moment of inertia Jy occupied by the water part of the annular flow was only slightly reduced from 5.11 to 4.94%.

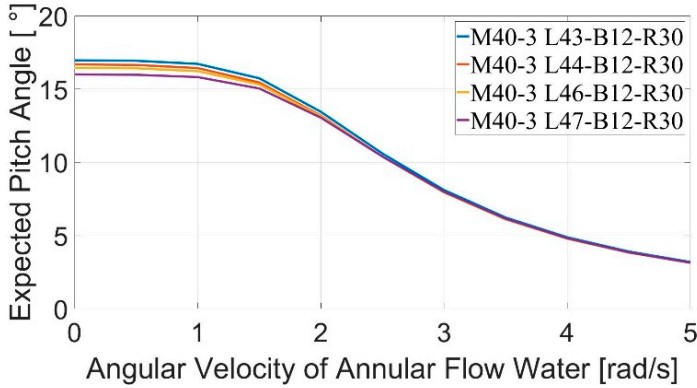

**Figure 15.** The expected pitch angle in the irregular wave upon changing the central cylinder height.

### 4.3.5. The Central Cylinder Wall Thickness

From Figure 16, we can see that, as the central cylinder wall thickness was increased from 0.01 to 0.2 m, when the angular velocity of annular flow was relatively low, the hydrodynamic performance worsened. At this time, the GM was increased from 1.23 to 2.71 m. When the angular velocity of the annular flow reached 3 rad/s, the hydrodynamic response tended to be the same, because the proportion of the moment of inertia Jy occupied by the water part of the annular flow only slightly increased from 4.70 to 5.10%. Therefore, the calculation of appropriate scale design parameters for annular flow presented in this section is universal.

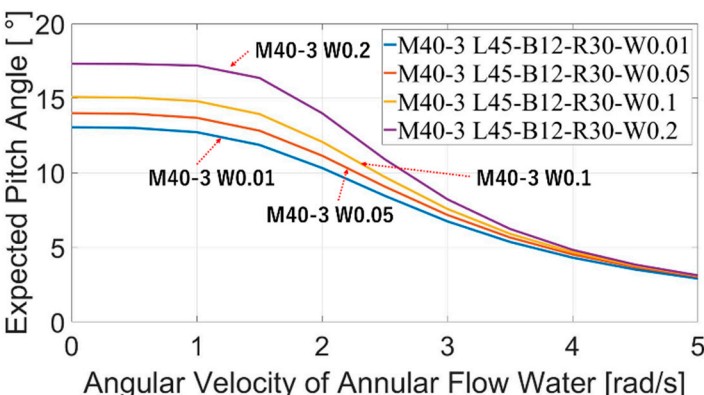

**Figure 16.** The expected pitch angle in the irregular wave upon changing the central cylinder wall thickness.

### 4.4. The Rotational Kinetic Energy

In this section, the engineering possibility of the FOWT from the viewpoint of energy consumption is discussed. The M40-3 case was selected, with a maximum electric power generated from wind energy of about $2 \times 10^7$ W. The rotational kinetic energy is calculated by Equation (15) and as shown in Figure 17.

$$E = \frac{1}{2}I\omega^2,$$

(15)

where $I$ is the rotational inertia, and $\omega$ is the angular velocity of annular flow water.

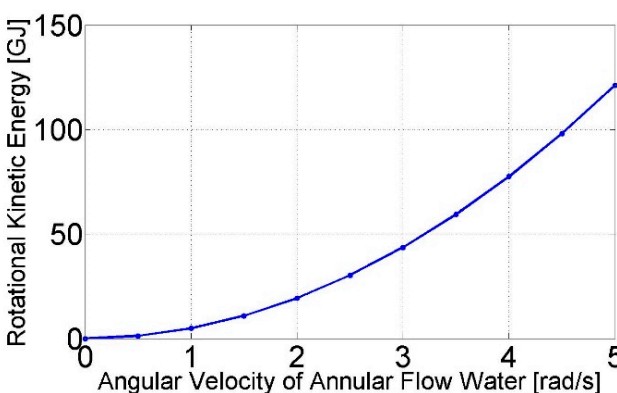

**Figure 17.** The rotational kinetic energy.

When the angular velocity of annular flow water was increased from 0 to 3 rad/s, resulting in a significant oscillating reduction effect, the energy required for water flow in the torus structure was 43.6 GJ, and the time taken to accelerate the flow water in the torus structure with the electricity generated by the FOWT was only 36 min. Of course, the calculation of the energy consumption should also include the energy needed to overcome the friction force to maintain the rotating motion. However, according to the calculation results of the rotational kinetic energy, the rotational angular velocity could be quickly increased to the required rotational angular velocity. Furthermore, in actual operation, the flow water in the torus structure would not necessarily have to accelerate from 0. Therefore, we can preliminarily conclude that the engineering possibility of this idea is very feasible.

Considering technical possibilities, despite frictionless phenomena such as the rotation of the suspended Moon around the Earth, superfluidity of helium with zero viscosity at ultralow temperatures [28], and biological superfluidity [29], they cannot be practically applied in engineering through technical means. The Maglev train is a more pragmatic solution, whereby an experimental Japanese maglev reached over 600 km/h in 2015. When applied to the M40-3 case in this paper, this would result in an angular velocity of 4.2 rad/s, which is greater than the required angular velocity of 3 rad/s.

## 5. Conclusions

For the novel idea of a cylinder-type FOWT using a torus structure with annular flow water, this paper mainly focused on hydrodynamic calculations, which led to the determination of the geometry of the torus structure and the required rotational angular velocity of the rotating part inside the torus structure. Moreover, the feasibility from the viewpoint of energy consumption was briefly analyzed in terms of realizability, which was indicated to be very high.

1.  The novel design using annular flow water in the torus structure as the spinning top was confirmed through experiments, and its influence on the hydrodynamic response was mainly by acting on the damping term as a damping force.
2.  According to the calculation results for the regular wave, it was revealed that, when the volume of annular flow water was rational, a large damping effect could be overwhelmingly confirmed, even for a relatively small angular velocity of annular flow water.
3.  According to the calculation results for the irregular wave, when the proportion of the moment of inertia Jy of annular flow water was about 5%, and the angular velocity of annular flow water was about 3 rad/s, and a significant oscillating suppression effect could be obtained.
4.  When a better oscillating suppression effect was obtained, the energy consumed by the annular flow water was not a large proportion of the power generation of the FOWT.

In the future, a complete dynamic analysis considering the aerodynamics of the part above the water plane will be conducted; developing an approach to minimize the friction in the torus structure is also a future direction worthy of research efforts.

**Author Contributions:** Conceptualization, M.M.; methodology, M.M.; software, M.M.; writing—review and editing, M.M.; supervision, M.M.; project administration, M.M.; validation, X.L.; formal analysis, X.L.; investigation, X.L.; data curation, X.L.; writing—original draft preparation, X.L.; writing—review and editing, X.L.; visualization, X.L. All authors have read and agreed to the published version of the manuscript.

**Funding:** This research received no external funding.

**Institutional Review Board Statement:** Not applicable.

**Informed Consent Statement:** Not applicable.

**Data Availability Statement:** Not applicable.

**Conflicts of Interest:** The authors declare no conflict of interest.

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
