# Peer review of "Engineering Possibility Studies of a Novel Cylinder-Type FOWT Using Torus Structure with Annular Flow"

_energies, doi:10.3390/en15134919_

Round 1

Reviewer 2 Report

Introduction

… MIT put forward the concept of FOWT for the first time in 1972. Please give a reference.

More references are needed. For example you can use for:

Spar type:

J. Jonkman and W. Musial, ‘‘Final Report, Subtask 2, The Offshore Code Comparison Collaboration (OC3), IEA Wind Task 23,’’ 2010.

Semi-submersible:

A. J. Goupee, B. J. Koo, K. F. Lambrakos and R. W. Kimball, ‘‘Model Tests for Three Floating Wind Turbine Concepts,’’ in the Offshore Technology Conference, 2012

Barge type:

Cermelli, C, Roddier, D, and Aubault, A (2009). “WindFloat: A Floating Foundation for Offshore Wind Turbines-Part II: Hydrodynamics Analysis," Proceedings of the ASME 2009 28th International Conference on Ocean, Offshore and Arctic Engineering. Volume 4: Ocean Engineering; Ocean Renewable Energy; Ocean Space Utilization, Parts A and B. Honolulu, Hawaii, USA. May 31–June 5, 2009. pp. 135-143. ASME. https://doi.org/10.1115/OMAE2009-79231.

TLP:

Mazarakos, Thomas P., Theodosis D. Tsaousis, Spyros A. Mavrakos, and Ioannis K. Chatjigeorgiou. 2022. "Analytical Investigation of Tension Loads Acting on a TLP Floating Wind Turbine" Journal of Marine Science and Engineering 10, no. 3: 318. https://doi.org/10.3390/jmse10030318

Wang, HF, and Fam, YH (2013). “Preliminary design of offshore turbine tension leg platform in the South China Sea,” J. Eng. Sci. Technol. Rev. 6 (3), 88-92.

Conceptual Design

Give some dimensions in Figure 1.

… The torus structure is very common in daily life. Please give some examples in the offshore engineering and references.

The space in the ring tube of the internal torus 79 is densely filled with fluid. Have you check sloshing phenomena? Make a comment.

What is the distance between the connection pieces? Have you check the structure due to structural analysis? Any regulations have been used to this connections?

… delaying its fatigue. Please give more details for the fatigue analysis, because in this idea you haven’t any moorings to take fatigue at the connection points.

Lines 86-88: Give some references on that.

Lines 96-100: Inside the torus for the heave motions of a structure we have moon pools, sloshing and wave trapping. Give some comments on this observations.

Formulation

… then the principle of linear superposition and the 118 maximum prediction theory for Longuet-Higgins statistics are used. Give some references.

For the eq. 2-5 please also make a sketch with the coordinate system used in this study.

Please use the same symbols in the equations and the manuscript (i.e. line 151 Φ, and eq. 6 Φ)

Give some references for eq. 9, 10, 11

Line 171: please change with exciting wave force

The Spinning Top

A sketch for this chapter is needed.

The Expected Value During Irregular Wave

Line 208-209: please give a reference.

Line 215: please give a reference for the PM Spectrum.

Eq. 20: Please explain all the symbols

Experimental Verification

Please give more details for the experiment. Dimensions, scale etc.

Table 1: the values of ωwater are from the model test. Please give an estimation for this values for a real structure.

Figure 4: In this figure only the 3 from the 4 values of the omega- damping are shown. Please make a bigger Figure, with better analysis.

The Object Parameters

Line 253. 7.5m/s. Is this value correct? For a 5 or 10 MW WT we need 11.4m/s. Please give a reference.

(BEM) theory: please give some references.

Give also a reference for eq. (22).

Table 2: Blade Length 100m. Is that value correct? This is the total diameter of the blades (Radius 50m?)?

Line 272: water depth. Is this water deep – intermediate or shallow due to kh (k: wave number, h: water depth) used in this study? Please make a comment.

Line 274: … The laboratory's existing fluid-solid coupling hydrodynamic CAE program: Please name the potential theory code used in this study and give a reference. More details for the model are needed (number of elements, number of points, ect.)

Line 275: please give a reference for Matlab (version also).

Figure 6: also the surge motion is needed because the structure is free floating. For the pitch what is the maximum angle (in degrees/wave amplitude). It is inside the regulations used in the offshore industry?

Line 309-310: for the wave spectrum you use Hs=9m (Hmax=16.74m) and Tp=20s. Is this set of parameters for a specific region? Usually for the offshore structures we use Tp near to 10s. Please give an explanation for this set of values for the PM spectrum.

Also some figures for the heave and surge motions are needed.

Figure 12: The maximum pitch angle is near to 28 degrees. The WT design was some limits near to 15. Make a comment for the High values of the Pitch angles.

References

More references are needed.

Round 2

Author Response

Thanks for your kind comments,  Please see the attachment of my reply.

Reviewer 2 Report

Introduction

More references are needed.

As a references (for the cases of Spar buoy, spar- submersible, TLP, barge-type) you can use the following:

Spar buoy:

J. Jonkman, D. Matha, “A quantitative comparison of the responses of three floating platform”, Proceedings of European Offshore Wind 2009 Conference and Exhibition, NREL/CP-500-46726, 2009.

Semisubmersible:

Butterfield, S, Musial, W, Jonkman, J, and Sclavounos, P (2007). “Engineering Challenges for Floating Offshore Wind Turbines,” Conference Paper NREL/CP-500-38776, September 2007.

Cermelli, C, Roddier, D, and Aubault, A (2009). “WindFloat: A Floating Foundation for Offshore Wind Turbines-Part II: Hydrodynamics Analysis," Proceedings of the ASME 2009 28th International Conference on Ocean, Offshore and Arctic Engineering. Volume 4: Ocean Engineering; Ocean Renewable Energy; Ocean Space Utilization, Parts A and B. Honolulu, Hawaii, USA. May 31–June 5, 2009. pp. 135-143. ASME. https://doi.org/10.1115/OMAE2009-79231

TLP:

P. D. Sclavounos, and E. N. Wayman, ‘‘Coupled dynamic modelling of floating wind turbine Systems,’’ in the OffshoreTechnology Conference, Texas, 2006.

Thomas P. Mazarakos, Theodosis D. Tsaousis, Spyros A. Mavrakos, Ioannis K. Chatjigeorgiou. “Analytical Investigation of Tension Loads Acting on a TLP Floating Wind Turbine,” J. Mar. Sci. Eng. 2022, 10(3), 318; https://doi.org/10.3390/jmse10030318

Bardge-type:

Guignier, L., Courbois, A., Mariani, R., Choisnet, T. 'Multibody Modelling of Floating Offshore Wind Turbine Foundation for Global Loads Analysis'. Proceedings of the Twenty-sixth International Ocean and Polar Engineering Conference, 26 June - 1 July, 2016, Rhodes, Greece

Ikoma. T., Tan, L., Moritsu, S. Aida, Y., Masuda, K.: 'Motion characteristics of a barge-type floating vertical-axis wind turbine with moonpools'. Ocean Engineering, Volume 230, 15 June 2021, 109006 https://doi.org/10.1016/j.oceaneng.2021.109006

Figure 1 (right):  How this structure mounted on the sea bed. Type of mooring lines (single point mooring, mooring line array, TLP, semi- taut)? Please give an explanation.

Eq. (1): please explain the terms included in mij (structure+wind turbine+torus).

Eq. (2): please give an explanation for J and *.

Line 194-195: … v is the wind speed at 19.5m. In general we have the value for the wind speed at 10m. How you find this value at 19.5 m (API regulations, DNV regulations). Please give an explanation.

Line 261: programs SS. More details are needed. Is this code commercial or in-house. It uses the pulsating Green’s function? This code removes the irregular frequencies? Please give more details.

Line 262: the number of nodes is 2581 (explain the type of elements, i.e. triangular). This value is the same for all the models used in this study? The mesh quality is checked? The model discretization converges? Give more details.

References

Reference 3: Is this reference correct?

Reference 21: Something is missing in reference 21.

Author Response

(The authors gave the same response as above.)

Round 3
